# Strength and Electrical Properties of Cementitious Composite with Integrated Carbon Nanotubes

**DOI:** 10.3390/ma16134771

**Published:** 2023-07-01

**Authors:** Anna Lushnikova, Olivier Plé, Yago De Souza Gomes, Xiaohui Jia, Wei Yang

**Affiliations:** 1Laboratoire Procédés Energie Batiment, Université Savoie Mont Blanc, Unité Mixte de Recherche du CNRS 5271, Institut National de l’Energie Solaire, 73000 Chambéry, France; anna.lushnikova@univ-smb.fr (A.L.); yago.de-souza-gomes@univ-smb.fr (Y.D.S.G.); xiaohui.jia@univ-smb.fr (X.J.); 2Key Laboratory of Building Safety and Energy Efficiency of the Ministry of Education, Hunan University, Changsha 410082, China; yangwei86@hnu.edu.cn; 3National Center for International Research Collaboration in Building Safety and Environment, Hunan University, Changsha 410082, China; 4College of Civil Engineering, Hunan University, Changsha 410082, China

**Keywords:** carbon nanotubes, tobermorite, cement mortar, electrical and mechanical properties, molecular dynamics

## Abstract

The main objective of this work was to study the effects of carbon nanotubes (CNTs) on the strength and electrical properties of cement mortar. Molecular dynamic simulations (MDSs) were carried out to determine the mechanical and electrical properties of a cementitious composite and its associated mechanisms. To model the atomic structure of a calcium silicate hydrate (C-S-H) gel, tobermorite 11Å was chosen. Single-walled carbon nanotubes (SWCNTs) embedded in a tobermorite structure were tested numerically. In particular, it was concluded that a piezoelectric effect can be effectively simulated by varying the concentration levels of carbon nanotubes. The deformation characteristics were analyzed by subjecting a sample to an electrical field of 250 MV/m in the z-direction in a simulation box. The results indicated a progressively stronger converse piezoelectric response with an increasing proportion of carbon nanotubes. Additionally, it was observed that the piezoelectric constant in the z-direction, denoted by d33, also increased correspondingly, thereby validating the potential for generating an electrical current during sample deformation. An innovative experiment was developed for the electrical characterization of a cementitious composite of carbon nanotubes. The results showed that the electrostatic current measurements exhibited a higher electric sensitivity for samples with a higher concentration of CNTs.

## 1. Introduction

Concrete, a very old material, has been widely used since the 19th century. It is the most commonly used material in the construction of civil engineering structures. Since climatic warming is a hard fact, its use is questioned because of its significant carbon footprint. Today, researchers are working to improve its physical and mechanical characteristics, as well as its durability, by mixing cement paste with different additive powders or nanoparticles. These nanoadded substances are generally used with superplasticizers. One of the most useful nanoadditives these days is carbon nanotubes (CNTs). The exceptional mechanical, electrical, and thermal properties of carbon nanotubes (CNTs) [1,2,3] make them beneficial for applications in various areas of civil engineering. The use of CNTs improves the mechanical properties of concretes, as reported in the literature [4,5]. However, objective of recent research is not only to focus on improving the mechanical properties but also to view CNTs as an adaptive and evolving material. Inducing the piezoresistive effect in cementitious materials to monitor structures can be achieved via the use of conductive fillers such as carbon nanofibers, carbon nanotubes, and graphene [6,7,8,9,10]. Azhari, F. et al. [8] considered cement-based sensors, including carbon nanofibers or carbon nanotubes, to change the resistivity, with good results for sensors with both nanotubes and nanofibers. Zhao, P. et al. [9] showed an improvement in the piezoelectric properties when carbon nanotubes were dispersed in cement-based composites in the range of 0 to 0.9 vol% (volume percentage). Gong et al. 2011 [10] created composites using Portland cement and PZT powders with the inclusion of modified CNTs ranging from 0 to 1.3 vol%. The best outcome was achieved when the concentration of the modified CNTs was 0.6 vol%. The results showed that the highest constant *d*_33_ (see Equation (2) below) was equal to 62 pC/N for a 0.3 vol% CNT. 

Electrical conductivity is one of the properties that the microstructure of cement cannot naturally develop. The resistivity of concrete varies from 105 Ω/mm to a range of 1012 Ω/mm between wet and dry concrete, respectively, changing from a semiconductor to an insulator in these two phases, thereby allowing evaporable water to be considered as having an important role in electrical conductivity. The ion concentration in water induces an ionic association that leads to the formation of C-S-H gel, consequently decreasing the mobility of the ions due to the electrical insulation layer in the cement grains. After modeling a concrete sample, the conductivity of the aqueous phase, facilitated by the dissolved ions, was significant. So, the resistivity was very low; however, this resistivity increased with the hydration of the concrete, indicating a proportionality between the evolution of resistivity and the evolution of strength. This variation in resistivity could help predict the performance and thus indicate the age factor of structures [1]. At this point, the modification of concrete to obtain electrical properties is very promising for structural health monitoring applications, for example. In this study, we combined experimental and numerical research to investigate the effects on mechanical and electrical properties caused by the modification of cement with carbon nanotubes. The first part of our study focuses on numerical simulations. In these simulations, the dynamics occur through discrete interactions; in our case, this occurs with atoms, which are called particles. The number of particles used in the simulations in this study varied depending on the number of carbon nanotubes incorporated into the tobermorite structure. These simulations were made possible by the ongoing development of the LAMMPS program for molecular modeling, which is a computational tool that continuously improves its ability to accurately represent the physical, chemical, and biological characteristics of systems composed of interacting molecules and atoms [11].

In addition, innovative experimental tests were conducted to measure the electrical currents in prismatic mortar samples of 40 × 40 × 80 mm with carbon nanotubes in weight percentages (wt.%) by weights of cement of 0.00 wt.%, 0.006 wt.%, and 0.018 wt.%. The effectiveness of CNTs in modifying the properties of concrete is largely determined by their level of dispersion [12]. There is a general consensus that standard mixing or manual mixing methods are insufficient for achieving the uniform dispersion of CNTs [13]. Consequently, various dispersion techniques, such as sonication and CNT functionalization, have been devised to achieve stable and uniform aqueous CNT dispersions [14]. It should be noted, however, that the CNT dispersion obtained through sonication tends to re-agglomerate over time, resulting in a decrease in the dispersion quality. For the experimental test, we utilized a Masterbatch CW2-45, which was produced by Arkema Group Co. To disperse the premix, we employed a high-speed bead mill mixer with 55% carboxymethylcellulose and 45% MWCNT. This Masterbatch can be used directly in the water needed to prepare the cement paste, which helps to avoid the coagulation of carbon nanotubes in the dispersion. This technology has been patented for use in cement systems [15].

Therefore, the aim of this paper was to focus on the modification of the cement paste structure via the addition of CNTs. To achieve this, we aimed to determine the atomic structure of the C-S-H/CNT composite and simulate it.

## 2. Molecular Dynamics Simulation

To elucidate the initiation of electrical conductivity and strength improvement in the cementitious composite with CNTs, an understanding of the interaction processes between nanoadditives and cement paste is needed. The electrical and mechanical properties of the cement paste containing CNTs were determined using a molecular dynamics study. The piezoelectric effect can be obtained using applied strains. Under compression or tension, the atoms change position in the crystal structure and generate a voltage. The polarization along the r-axis can be obtained via the following process: (1)$$P_{r}=\frac{\sum_{i}^{N}q_{i}r_{i}}{V},$$
where *q_i_* is the charge of the *i*th atom, *r_i_* is the *r*-coordinate of the *i*th atom, *N* is the number of atoms, and *V* is volume. 

When an electric field is applied to generate deformations, the piezoelectric effect is simulated. It can be represented by matrices [11] to obtain the strain resulting from the elastic, thermal, electric, and magnetic phenomena. The tensors for variables such as the stress $\sigma_{kl}$, electric field $E_{k}$, magnetic field $H_{l}$, and temperature differential ∆T can be represented as follows:(2)$$\varepsilon_{ij}=S_{ijkl}\sigma_{kl}+d_{kij}^{T}E_{k}+q_{lij}H_{l}+\alpha_{ij}∆T$$
where $S_{ijkl}$ is the elastic compliance, $d_{kij}$ is piezo-electric, $q_{lij}$ is piezo-magnetic, and $\alpha_{ij}$ is the thermal expansion tensor. 

Apart from the aforementioned equation, we can represent the effects of interest in this study using another equation that eliminates the magnetic effect. This can be expressed as:(3)$$D_{i}=d_{ijk}\sigma_{jk}+\epsilon_{ik}E_{k}+p_{i}∆T$$
where $D_{i}$ is the electric displacement field, $d_{ijk}$ is the piezo-electric tensor, $\epsilon_{ik}\;$is the material electric permittivity, and $p_{i}$ is the pyro-electric tensor.

For this study, we selected tobermorite as the main constituent of cement after hydration, as it has a similar structure to that of the C-S-H gel. Tobermorite is a crystalline and natural mineral with the chemical formula Ca_5_Si_6_0_16_(OH)_2_4H_2_O. There are different subspecies of tobermorite categorized by their interlayer spacing, with tobermorite 11Å being the most commonly used model to describe cement paste in MD simulations [16,17,18]. Its initial structure is monoclinic and has vectors *a* = 6.735Å, *b* = 7.385Å, and *c* = 22.487Å, and angles between vectors *α* = 90°, *β* = 90°, and *γ* = 123.25°, as determined by Merlino in 1999 [19].

To create the initial configuration system, a Python code was utilized. The triclinic unit cell basis can be expressed in a matrix format using the following formula:(4)$$\left(a\;b\;c\right)=\left(\begin{array}{ccc} a_{x} & b_{x} & c_{x}\\ 0 & b_{y} & c_{y}\\ 0 & 0 & c_{z} \end{array}\right)$$
(5)$$a_{x}=A$$
(6)$$b_{x}=B.Â=B\cos\gamma$$
(7)$$b_{y}=\left|Â\;x\;B\right|=B\sin\gamma =\sqrt{B^{2}-b_{x}^{2}}$$
(8)$$c_{x}=C.Â=C\cos\beta$$
(9)$$c_{y}=C.\left(A\;x\;B\right)\;x\;Â=\frac{\left(B.C-b_{x}c_{x}\right)}{b_{y}}$$
(10)$$c_{z}=\left|C.\left(A\;x\;B\right)\right|=\sqrt{C^{2}-c_{x}^{2}-c_{y}^{2}}$$

The vector format was used to include the fractional positions and the positions were extended in the three cartesian dimensions using three loops based on the system size to create a supercell, as shown in Figure 1. We employed tobermorite 11Å supercells of size 5 × 5 × 3(Å), containing a total of 9683 to 9899 atoms in this project.

We opted for a single-wall carbon nanotube, specifically the armchair (3,3)-type with a diameter of 4.073Å and length of around 70Å, due to its ideal diameter-to-insertion ratio within the tobermorite structure. The nanotubes were inserted into the tobermorite structure along the z-axis (Figure 1) after establishing a vacuum in the structure. In the context of this investigation, the word “vacuum” denotes the empty space generated to accommodate the nanotube insertion into the tobermorite structure while preserving the system’s neutral charge. This was accomplished by refraining from allocating unit cells to a specific location in the xy plane, resulting in the absence of atoms within the simulation box. In Figure 2, to create a void in the xy plane and along the z-direction for the 5 × 5 × 3 supercell to insert one CNT, unit cells U331, U332, and U333 were not considered in the code. As a result, the code created unit cells around the nanotube. When the unit cell’s geometry, particularly its cross-sectional dimensions, closely matches the targeted hole diameter, the ensuing vacuum within the monoclinic-shaped tobermorite 11Å exhibits a cylindrical appearance after the minimization procedure, as illustrated in Figure 2. Therefore, it could be advantageous to consider the dimensions of the unit cell during the creation of the hole to obtain the desired shape of the void. There are alternative approaches to incorporating CNTs into tobermorite, apart from the method mentioned previously. For example, in a study by Eftekhari M. et al. [20], a cylindrical diameter of a hole, which was initially zero, was incrementally increased using the “fix ident cylinder” command of LAMMPS [21]. Generating a vacuum is also possible by removing the less stable atoms while keeping the molecular neutral following the verification of the charge [22]. Plassard C. et al. [23] noted the absence of silica tetrahedral in tobermorite like C-S-H and accordingly identified the silicon atoms with lower stability within the silicon chains. By eliminating the less stable silicon atoms, the system’s charge can remain neutral while altering the geometry of the silica tetrahedron to a silica trigonal pyramidal structure.

To govern the interaction between the carbon atoms of CNT, we adopted the Tersoff potential [24]. The cumulative energy associated with the interatomic forces between all the atoms within the tobermorite structure is derived from the summation of all the various types of atomic interactions. Such interactions between the atoms of tobermorite and CNT are described in numerous studies, such as in [25,26]. 

To examine the incidence of the concentration of CNT in the cement paste, systems without CNT and one, two, and three CNTs that yield 0.00%, 1.22%, 2.55%, and 4.00% of CNTs by weight (wt.%) of tobermorite were prepared (Figure 3). To determine the concentrations, one can calculate the product of the total number of atoms present in the system and their respective atomic masses.

The simulation employed periodic boundary conditions to simulate a larger system and prevent the boundary effect. After minimization, an NPT ensemble was utilized with a time step of 1 femtosecond at a temperature of 300 K and zero pressure to regulate the temperature and pressure of the simulation box.

## 3. Experimental Program

### 3.1. Materials

#### 3.1.1. Carbon Nanotubes

Multi-wall carbon nanotubes (MWCNT) from Arkema company were used in this study. The product used for this experiment was Graphistrength^®^ CW 2-45, which is a masterbatch with a high concentration of 45% by weight of pre-dispersed carbon nanotubes Graphistrength^®^ C100 (Figure 4) with the typical properties presented in Table 1, perfectly dispersed in carboxyl methyl cellulose (CMC).

Graphistrength^®^ CW 2-45 is presented in the form of pellets with the key characteristics given in Table 2.

#### 3.1.2. Cement, Sand, and Mortar

PERFORMAT® CEM I 52,5 N CE PM-CP2 NF was used to produce the mortar. Table 3 summarises the composition of clinker and some physical properties of the cement used in this work, provided by the Vicat group, France. 

Standard sand (ISO standard sand) was certified in accordance with EN 196-1, which is siliceous, and its finest fractions with the grading given in Table 4 were used.

Mortar mix with 0.006 wt.% of MWCNTs was chosen as a principle mix based on the results of [28] that show the best concentration of CNT in cement concrete using the same carbon nanotubes from Arkema. We hypothesized that this represents 1.22% of the CNTs by weight of tobermorite. To obtain the proportionality of the nanotube concentration compared to the simulation and obtain additional results, we prepared a mix with 0.018 wt.% of MWCNTs that represent 4.00% of CNTs by weight of tobermorite.

Thus, mortar mixes with 0.00 wt.%, 0.006 wt.%, and 0.018 wt.% of MWCNTs were prepared. The samples were mixed using a laboratory mixer. 

We used 40 × 40 × 160 cm^3^ prismatic molds to prepare the samples with two 2D inox meshes 2 × 2 cm placed in the normal axis 2 cm apart from each other to simulate the electrodes of the capacitor (see Figure 5 and Figure 6) and homogenizing via a vibrating table. Table 5 shows the mixing proportions. After curing for 28 days, the samples were cut into two parts to obtain 6 of each concentration of MWCNTs of size 40 × 40 × 80 cm^3^ (see Figure 5). 

### 3.2. Testing Procedure

The samples were heated by an infra-red lamp to simulate the sun, and the corresponding electrostatic current was measured with a Keithley model 6482 dual-channel picoammeter. The temperatures were measured using a thermal camera FLIR Systems AB model 60b × 1.0 (Figure 5). As shown in Figure 5, the samples with and without MWCNTs look identical. Only a scanning electron microscope (SEM) analysis can be used to differentiate them.

The Keithley model 6482 dual-channel picoammeter was connected to the two electrodes of the capacitor (Figure 6). The leakage current was measured when the sample was subjected to a temperature field.

## 4. Results and Discussion

### 4.1. Dynamic Molecular Study

The Young’s modulus of the built model was evaluated and compared with previous research using Table 6 and Table 7. The system was equilibrated before undergoing deformation via a uniaxial tension by fixing one side of the structure and stretching the other side at a strain rate of 0.00001/fs. The initial length of the simulation box in the z-direction, as shown in Figure 3, was saved before this process.

After carrying out separate tensile tests on tobermorite 11Å and CNT (3,3), the combination of tobermorite 11Å + CNT was subjected to the same test to determine the effect of CNT concentration on the tensile resistance of tobermorite 11Å. Table 8 and Figure 7 present the results for Young’s modulus, tensile strength, and stress evolution due to applied strain for the tobermorite-CNT models. The initial stress peak at 0.1 strain (Figure 7) indicates bond breaking in the tobermorite atoms (Figure 8b), followed by an increase in stress as only the CNTs were strained (Figure 8c) until the CNTs ruptured at around 0.25 strain (Figure 8d).

Subsequently, the piezoelectric effect was simulated. The system was first equilibrated, and then the lengths of the simulation box were monitored while an external electric field of 250 MV/m was applied in the z-direction (as shown in Figure 3). The evolution of the box length in the z-direction for the four systems is compared and presented in Figure 9.

The strain tensor $\varepsilon_{ij}$ was determined between simulation steps 4000 and 10,000, resulting from an electric field of 250 MV/m applied in the z-direction (refer to Figure 3). As there was no external stress or temperature change, Equation (2) for the converse piezoelectric effect in a triclinic structure could be expressed as follows:(11)$$\varepsilon_{ij}=d_{ijk}^{T}E_{k}$$

This can be solved for $d_{33}$, yielding: (12)$$\varepsilon_{3}=d_{33}\;*\;250\;MV/m$$

The piezoelectric constant calculated for each of the samples is shown in Table 9.

The tobermorite 11Å structure alone displays a higher ultimate tensile strength than the models Tob-CNT (1.22 wt.%) and Tob-CNT (2.55 wt.%). However, the Tob-CNT model (4.00 wt.%) has the highest tensile strength of 14.27 GPa and the highest Young’s modulus of 165.46 GPa, as shown in Table 8. It is observed that the models Tob-CNT (1.22 wt.%) and Tob-CNT (2.55 wt.%) exhibit a decreased maximum tensile strength. In all cases where CNTs were incorporated into tobermorite, a higher Young’s modulus was observed. The initial stress peak, which is associated with the collapse of the tobermorite structure, occurs at a strain of 0.1. Figure 9 shows multiple stress peaks, which indicate that collapse occurs for strains ranging from 0.25 to 0.3 when CNTs are present. Despite a discontinuous collapse in the modeling when translating reality, cracking should not be separated. This numerical result can be interpreted as a discontinuity in the overall strain. Under the application of an electric field, the samples follow displacements, and the sample oriented in the same direction (z) of the electric field is analyzed. The piezoelectric constant is aligned in this direction, and the highest strain of 0.002155 was observed in Tob-CNT (2.55 wt.%). The samples with 1.22 wt.% and 4.00 wt.% displayed similar results. Without CNTs, the response to the applied electrical field is weak. By taking into account these numerical considerations, it should be experimentally possible to cause the deformation of a mortar sample containing CNT by a strain field (temperature field, for example), and thus generate an electric field. That is what we have tested in the following section.

### 4.2. Experimental Study

A total of 137 measurements of current were carried out on 6 samples of size 40 × 40 × 80 cm^3^ (see Figure 5) with 0.00 wt.% of MWCNTs. For 0.006 wt.% and 0.018 wt.% of MWCNTs, 6 samples were prepared and 109 measurements of current were carried out on each mixture. A solar lamp was used to simulate these materials. It can produce UVA or UVA-UVB. This infrared lamp was chosen to simulate sun exposure. The temperatures were measured using a thermal camera (Figure 5). The values of the current obtained from the experimental test in ηA with respect to the temperature for all the samples are displayed in Figure 10. Only the measurements where the temperature was stabilized at 20 °C (ambient temperature), 30 °C, 35 °C, 45 °C, and 50 °C were retained in the range of ±2 °C. The results show great dispersion (see Figure 10) because electrical measurements at such low levels are very sensitive to the environment. An electrically neutral enclosure, controlled by temperature and humidity, should have been used. This is planned as the next step of this research. Nevertheless, the tendencies (see Figure 10) show that the current increases for the same temperature when the percentage of wt.% of MWCNTs increases in the mixture, which was expected, given the dynamic molecular study.

As shown in Equation (2), the strain (ε) of a material can be expressed as the change in temperature (∆T) multiplied by the coefficient of thermal expansion (α). For mortar, the coefficient of thermal expansion is in the order of $10\times 10^{-6}$/°C [36]. Table 10 uses α∆T to express the increment of the measured current concerning the strain by averaging the samples that have the same CNT concentration. The average electrostatic current increment for a strain of $10\times 10^{-5}$ and ∆T= 10 °C increases between 1.28 (ηA) and 1.51 (ηA) when the percentage increases between 0.000 wt.% and 0.018 wt.%.

Dynamic molecular and experimental studies demonstrated that the piezoelectric effect can be obtained in a mortar containing MWCNTs. As in certain ceramics, piezoelectricity is the electric charge that accumulates in solid materials in response to applied stresses and reciprocally. This electricity, resulting from the displacement caused by an increase in temperature, can be capped. The piezoelectric phenomenon arises due to the interdependent relationship between the mechanical and electrical properties of materials. By including MWCNTs in a mortar, we can achieve a piezoelectric effect. This piezoelectric effect can be used as a sensor to analyze the field of stress detection caused by loads or temperature.

## 5. Conclusions

This study examined how the inclusion of carbon nanotubes affects the electrical properties of cement composites. To investigate this, a molecular dynamics simulation was performed using tobermorite 11Å as a representative of calcium silicate hydrate gel, combined with armchair (3,3)-type single-wall carbon nanotubes. Thus, numerical simulations of tobermorite mixtures with different percentages of carbon nanotubes were prepared. First, the molecular dynamics study showed that the piezoelectric effect can be simulated for four different concentrations. The deformation in the simulation box was then analyzed by subjecting it to an electrical field of 250 MV/m in the z-direction, The results showed a stronger converse piezoelectric response when the percentage of carbon nanotubes was increased. The deducted piezoelectric constant in the z-direction d_33_ increased in the same manner, proving in theory that current can be generated when samples are subjected to strains. 

The experimental investigation was conducted using samples of mortar reinforced with different amounts (0.00 wt.%, 0.006 wt.%, 0.018 wt.%) of tangled multi-walled carbon nanotubes from Arkema. The samples were heated by an infra-red lamp to simulate the sun, and the corresponding electrostatic current was measured with a Keithley model 6482 dual-channel picoammeter. In parallel, the temperatures were measured using a thermal camera with a precision of ±2 °C. The results showed that electrostatic current measurements exhibited a higher electric sensitivity for samples with a higher concentration of CNTs. These experimental results are very promising when compared with molecular dynamics studies, and such effects can be used for analyses of stress detection caused by loads or temperatures in concrete structural elements.

## Figures and Tables

**Figure 1 materials-16-04771-f001:**
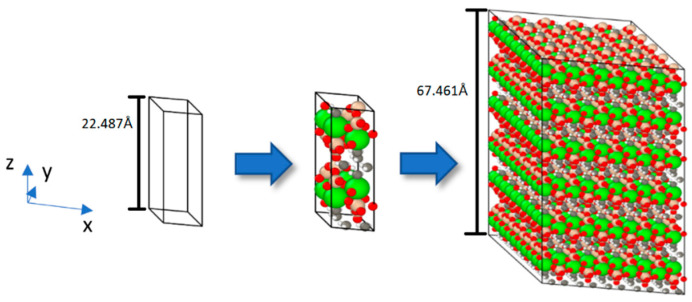
The process used to generate the configuration of the tobermorite 11Å system.

**Figure 2 materials-16-04771-f002:**
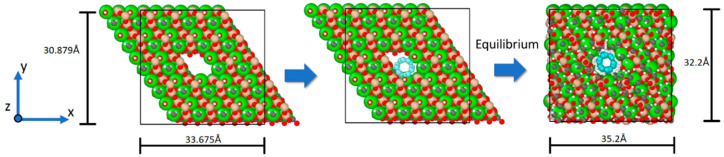
The procedural steps for incorporating a CNT within the tobermorite 11Å unit cell.

**Figure 3 materials-16-04771-f003:**
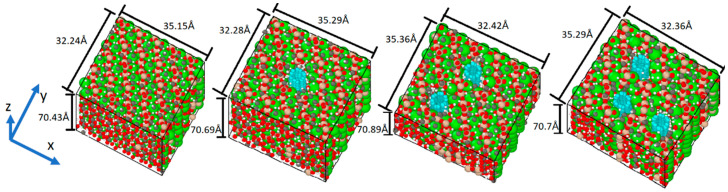
Four different systems comprising the minimized CNT-reinforced tobermorite structures were employed in the simulations.

**Figure 4 materials-16-04771-f004:**
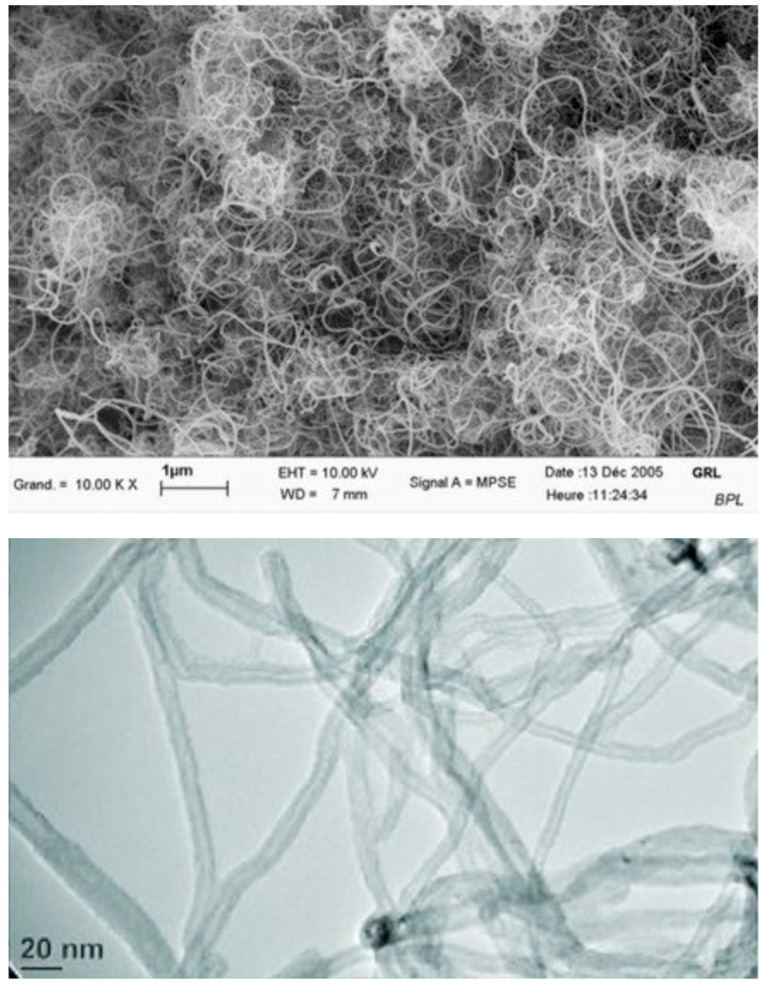
Graphistrength^®^ C100 at different scales [27].

**Figure 5 materials-16-04771-f005:**
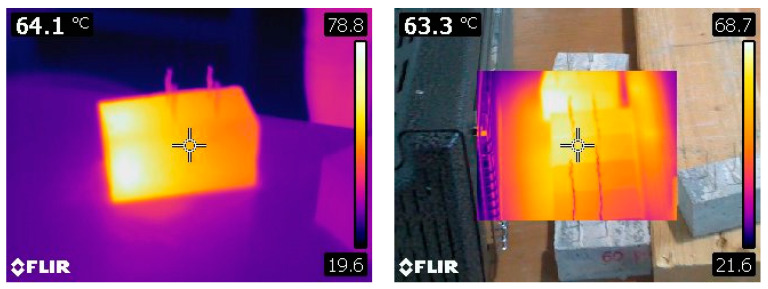
Samples 40 × 40 × 80 cm^3^ and temperature measurement by a thermal camera. Picture samples with and without MWCNTs.

**Figure 6 materials-16-04771-f006:**
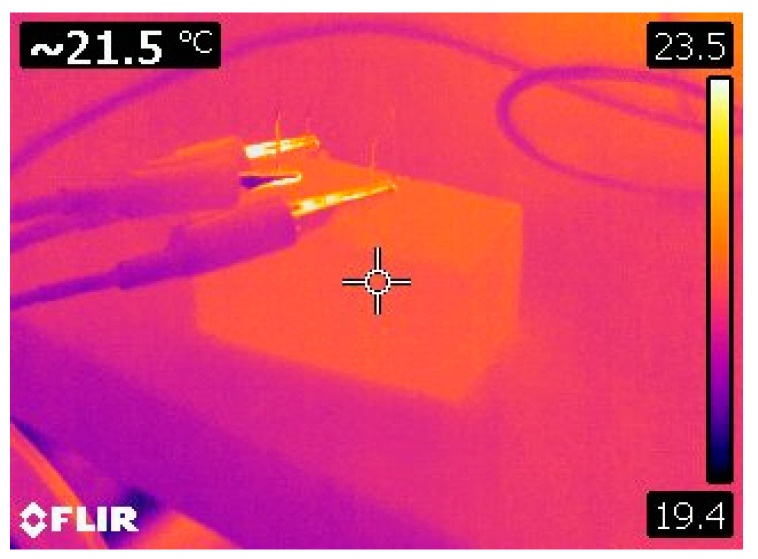
Samples 40 × 40 × 80 cm^3^ with MWCNTs and electric field measurement.

**Figure 7 materials-16-04771-f007:**
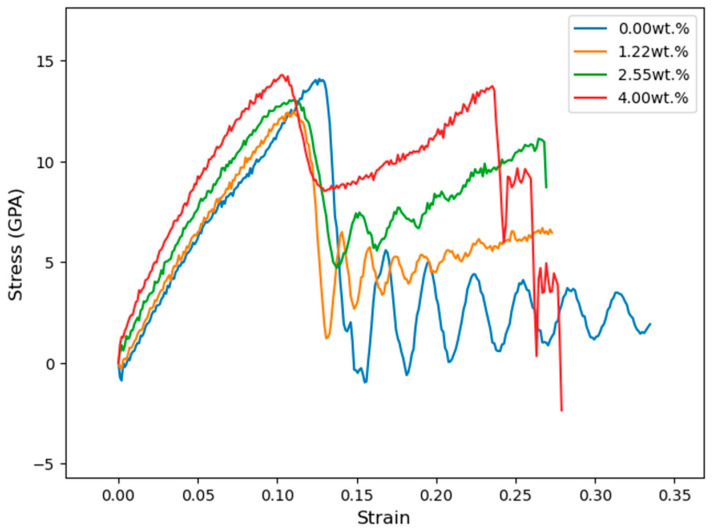
Tensile stress vs. strain for the tobermorite-CNT sample in the z-orientation.

**Figure 8 materials-16-04771-f008:**
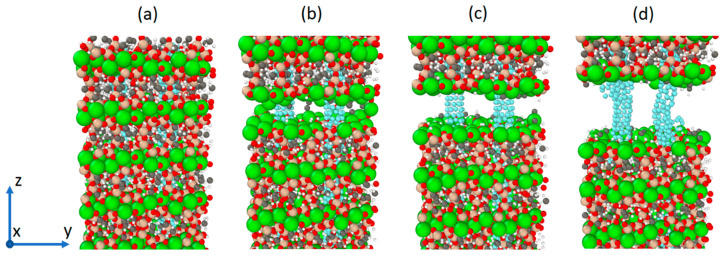
Tobermorite 11Å containing 2.55 wt.% of CNTs under uniaxial tension in the longitudinal direction is shown in four different stages: (**a**) at zero strain, (**b**) at a strain of 0.125, (**c**) at a strain of 0.15, and (**d**) at a strain of 0.25.

**Figure 9 materials-16-04771-f009:**
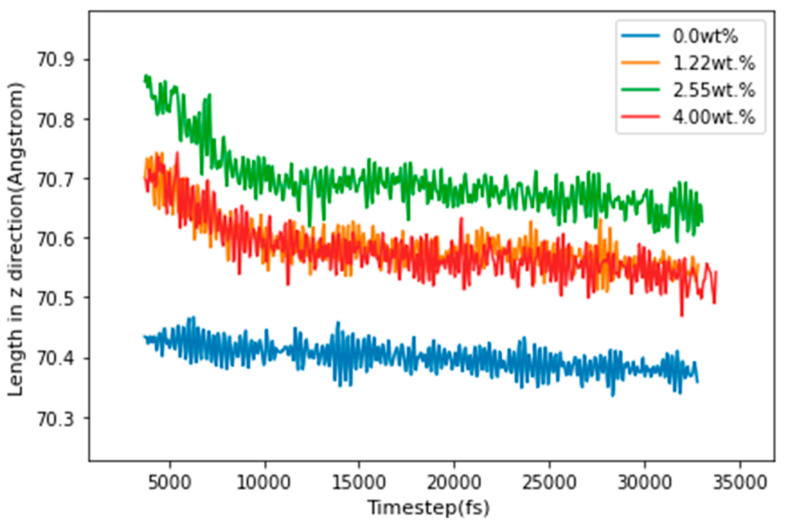
Simulation of the changing of cell size in z-orientation with varying CNT concentrations under an external electric field.

**Figure 10 materials-16-04771-f010:**
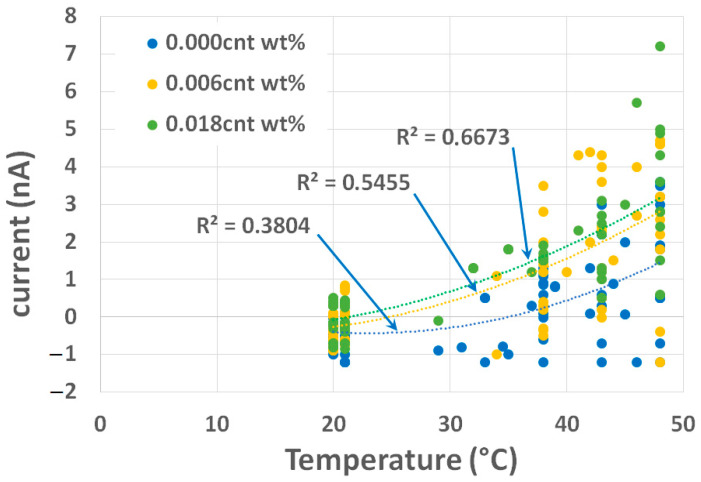
Measured values of current (nano-ampere) vs. temperature (°C) for mortar samples with three different concentrations of carbon nanotubes.

**Table 1 materials-16-04771-t001:** Properties of multi-walled carbon nanotubes (MWCNTs).

Description		MWCNT Graphistrength C100
Production process		Catalytic chemical vapor deposition (CCVD)
Powder characteristics	Apparent density	50–150 kg/m^3^
	Mean agglomerate size	200–500 μm
	Weight loss at 105 °C	<1%
MWNT characteristics	C content	>90 wt.%
	Free amorphous carbon	No detectable (SEM/TEM)
	Mean number of walls	5–15 nm
	Outer mean diameter	10–15 nm
	Length	0.1–10 μm

**Table 2 materials-16-04771-t002:** Main characteristics of Graphistrength^®^ CW 2-45.

Characteristic	Unit	Typical Value
Aspect		Black Pellets
MWCNT with purity > 90% content	wt.%	55
CMC content	wt.%	45

**Table 3 materials-16-04771-t003:** Composition of clinker and some physical properties of source cement.

Mineralogical phases of clinker	C_3_S (%)	60.0
	C_2_S (%)	15.5
	C_3_A (%)	7.1
	C_4_AF (%)	10.8
Physical characteristics	Density (g/cm^3^)	3.17
	Blaine number (cm^2^/g)	4150
	Setting start time (min)	150
	equivalent diameter (μm)	11.8

**Table 4 materials-16-04771-t004:** Grading of the source sand.

Square Mesh Size (mm)	Cumulative Retained (%)
0.08	99 ± 1
0.16	87 ± 5
0.50	67 ± 5
1.00	33 ± 5
1.60	7 ± 5
2.00	0

**Table 5 materials-16-04771-t005:** Materials proportion for mortar mixing.

Reference	MWCNTs Content (g)	Cement (g)	Water (g)	Sand (g)
Control	0.0	450	225	1350
CEM/CNT-0.006	2.7	450	225	1350
CEM/CNT-0.018	8.1	450	225	1350

**Table 6 materials-16-04771-t006:** Results of the computed Young’s modulus of tobermorite 11Å.

Tobermorite 11Å	Young’s Modulus (GPa)	Method
Present work	114.50	Molecular Dynamics
Dharmawardhana et al. [29]	103.25	Molecular Dynamics
Lushnikova [30]	78.39	Molecular Dynamics
R. Shahsavari et al. [31]	82.82	Ab initio

**Table 7 materials-16-04771-t007:** Results of determinate Young’s modulus of SWCNT.

SWCNT	Young’s Modulus (TPa)	Method
Present work	1.17	Molecular Dynamics
Wang et al. [32]	1.28–1.48	Molecular Dynamics
Lushnikova [33]	0.510	Molecular Dynamics
X. Lu, Z. Hu [34]	0.989–1.058	Finite Element Analysis
P Subba Rao [35]	1.22–1.28	Finite Element Analysis

**Table 8 materials-16-04771-t008:** Results of Young’s modulus and tensile strength values of tobermorite-CNT samples.

Model	Young’s Modulus (GPa)	Tensile Strength (GPa)
Tobermorite 11Å	114.05	14.07
Tob-CNT(1.22 wt.%)	139.98	12.49
Tob-CNT(2.55 wt.%)	148.76	13.03
Tob-CNT(4.00 wt.%)	165.46	14.27

**Table 9 materials-16-04771-t009:** Piezoelectric constant $d_{33}$ for each of the four samples.

Sample	0.00 wt.%	1.22 wt.%	2.55 wt.%	4.00 wt.%
*d*_33_ (*pC/N*)	1.12	6.48	8.62	6.14

**Table 10 materials-16-04771-t010:** Average electrostatic current increment for mortar samples with three different concentrations of carbon nanotubes for a strain of $10\times 10^{-5}$ and ∆T = 10 °C.

Concentration	0.00 wt.%	0.006 wt.%	0.018 wt.%
Δ*i* (*ηA*)	1.28	1.38	1.51

## Data Availability

All relevant data are contained in the present manuscript.

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
