# Peer review of "Strength and Electrical Properties of Cementitious Composite with Integrated Carbon Nanotubes"

_materials, 2023, doi:10.3390/ma16134771_

Round 1

Reviewer 1 Report

I find that the paper entitled: "Strength and electrical properties of cementitious composite with integrated carbon nanotubes" is an interesting.

In this paper authors were to study the effects of carbon nanotubes (CNT) on the strength and electrical properties of cement mortar. In support of this, a molecular dynamic simulation (MDS) was performed using Tobermorite 11Å as a representative of Calcium Silicate Hydrates gel, combined with Armchair (3,3) type single-wall carbon nanotubes. Thus, the models of Tobermorite mixes with percentage of carbon nanotubes were prepared.

Experimental investigation was conducted using samples of mortar reinforced with different amounts of tangled multi-walled carbon nanotubes from Arkema (mortar mixes with 0.00 wt.%, 0.006 wt.%, 0.018 wt.% of MWCNTs). Samples were heated by an infra-red lamp to simulate the sun, and the corresponding electrostatic current was measured with a picoammeter Keithley model 6482 Dual-channel. The authors presented the results which showed that electrostatic current measurements exhibited a higher electric sensitivity for the samples with a higher concentration of CNTs (0.018 weight percentage). The average electrostatic current increment, for a strain of 10 ? 10−5 and ∆?= 10°C, increases between 1.28 (??) and 1.51 (??) when the percentage increases between 0.000 wt% and 0.018 wt%.

The authors stated that these experimental results comparing with the dynamics molecular study are very promising and such effects can be used for field analyzes of stresses detection caused by loads or temperatures in concrete structural elements.

I suggest the authors just shorten the Conclusions and transfer part of the analysis they presented in the Conclusions in the part interpretation of the results. In conclusion there is redundant data.

In addition, I ask the authors to corection technical errors:

- Chapter 3.1.1. Cement, Sand and Mortar should be 3.1.2. Cement, Sand and Mortar,

- Figures 7, 8 and 9 are in italics, but the other pictures are not.

Accordingly, I recommend Accept after minor revision (corrections to minor methodological errors and text editing).

Reviewer 2 Report

This study presents the strength and electrical properties of cementitious composite with integrated carbon nanotubes. Comments are listed as follows:

1. It will be interesting to see a picture with the blank and the sample with carbon nanotubes. It can be a photo or an SEM image.

2. What thermal camera was used in the case? 

3. In temperature measurement, it is interesting to see the sample with carbon nanotubes cooperating with a blank sample. The same thing happened in the electric field measurement.

4.  I don't see the mortar and mortar with MWCNT recipe.

5. The bibliography need to be according with the journal requirment.

Reviewer 3 Report

Authors conducted experimental and numerical studies to investigate the contribution of carbon nanotubes (CNTs) to the mechanical and electrical properties of cement mortars. In general, the manuscript is well written. However, there must be a more rigorous interpretation and scientific explanations of the mechanisms of CNTs affecting various properties of cementitious composites. The following comments must be addressed to add value to the manuscript:

Abstract: Please summarize the main points and avoid unnecessary parts. I believe adding some of the most critical quantitative results to the Abstract would be appealing to the readers.

Introduction is somewhat poor (incomplete) given the scope of this study. This must be improved. The most relevant knowledge attained in previous studies should be considered in this manuscript and presented, as a summary. For example, please discuss the challenges associated with utilizing CNTs such as dispersion and bonding issues, and what methods are used to overcome these issues (e.g., dispersion methods, CNT surface treatment or functionalization, etc.). If CNTs lack proper dispersion, they introduce defects to the cement matrix, resulting in the degradation of its properties. This reference might be helpful to discuss: "Mechanical properties of carbon-nanotube-reinforced cementitious materials: database and statistical analysis."

1. Introduction: Authors stated that Concrete is not only a mechanically strong material; concrete becomes presently adaptive evolutionary. This statement is not true. Concrete is weak in tension. Fibers are added to concrete to increase the tensile strength. Please revise this. 

1. Introduction: “… cement-based sensors including carbon nanofibers or carbon nanotubes are supposed changing in resistivity with good results for sensors with both, nanotubes and nanofibers.” This is only true in the case of well-dispersed CNTs, and above their percolation threshold. Also, the effects of surfactants can be discussed. 

1. Introduction: “… with carbon nanotubes in weight percentages (wt.%) of 0.00wt.%, 0.006wt.% and 0.018wt.%.” Please specify based on the weight percent of what material.

2. Molecular Dynamics Simulation; Page 3: “We employed Tobermorite 11Å supercells of size 5?5?3, containing a total of 9683 to 9899 atoms, in this project.” Please add the unit for the size dimensions. Also, there is a syntax error (? should be replaced with a cross).

3. Experimental Program; Page 6: “Mortar mix with 0.006 wt. % of MWCNTs was chosen as a principle mix based on the results of [24] that show the best concentration of CNT in cement concrete …” 0.006wt% CNT content based on what? If it is based on the cement weight percentage, it seems very low content. The optimum CNT contents for improving various properties of cementitious materials were discussed in previous studies. 

Table 5: Please mention CNT contents in mass (g) like the other materials in this table.

4. Results and Discussion; Table 8: The tensile strength of the composite material degraded against the control (without adding CNTs) for 1.22wt% and 2.55wt%. Generally, the mechanical properties increase up to certain CNT content, beyond which it degrades due to dispersion issues. Please explain the reason for such a trend in molecular dynamic results.
